

# Implications of the SNHG10/miR-665/RASSF5/NF-κB pathway in dihydromyricetin-mediated ischemic stroke protection

Qi Zeng[1,2], Yan Xiao[3], Xueliang Zeng[4] and Hai Xiao[2,5]

[1] Department of Ultrasound, First Affiliated Hospital of Gannan Medical University, Ganzhou, China
[2] Key Laboratory of Prevention and Treatment of Cardiovascular and Cerebrovascular Diseases of Ministry of Education, Gannan Medical University, Ganzhou, China
[3] Department of Cardiovasology, First Affiliated Hospital of Gannan Medical University, Ganzhou, China
[4] Department of Pharmacology, First Affiliated Hospital of Gannan Medical University, Ganzhou, China
[5] Department of Pathology, First Affiliated Hospital of Gannan Medical University, Ganzhou, China

Corresponding author
Hai Xiao, 13576675898@163.com

## ABSTRACT

Ischemic stroke (IS) remains a leading cause of disability and mortality worldwide, and inflammation and oxidative stress play significant roles in its pathogenesis. This study investigates the effects of dihydromyricetin (DHM) on IS using RT-qPCR and western blot with SH-SY5Y cells, focusing on its effects on the small nucleolar RNA host gene 10 (SNHG10)/microRNA (miR)-665/Ras association domain family member 5 (RASSF5) axis and nuclear factor-kappa B (NF-κB) signaling. In addition, the effects of the SNHG10/miR-665/RASSF5 axis on SH-SY5Y cell activity, apoptosis, oxidative stress, and inflammatory markers were analyzed using 3-(4,5-dimethylthiazol-2-yl)-2,5-diphenyltetrazolium bromide, flow cytometry, and enzyme-linked immunosorbent assays. Our results showed that, in response to oxygen-glucose deprivation/reperfusion (OGD/R), DHM treatment improved cell viability, reduced apoptosis, and attenuated neuroinflammation and oxidative stress in a dose-dependent manner ($p < 0.05$). Interestingly, lncRNA SNHG10 was overexpressed during OGD/R and suppressed by DHM. Through bioinformatics analysis and experimental validation, we identified miR-665 as a direct target of SNHG10 and RASSF5 as a direct target of miR-665. The protective effect of DHM against OGD/R injury was partially reversed by SNHG10 overexpression and further enhanced by co-transfection with the miR-665 mimic and si-RASSF5 ($p < 0.05$). This study identifies a novel mechanism of DHM against IS, which may act via modulation of the SNHG10/miR-665/RASSF5 axis and inactivation of NF-κB signaling, and offers a promising therapeutic target for IS.

## INTRODUCTION

Ischemic stroke (IS) is the leading type of stroke, characterized by rapid neuronal death due to oxygen and nutrient deprivation (*Ferreira-Atuesta et al., 2021*; *Tsao et al., 2021*). IS pathology includes oxidative stress, neuroinflammation, and apoptosis (*Koutsaliaris et*

*al., 2022*). Therefore, therapeutic strategies that simultaneously target these pathological features were the focus of this study.

Increasing evidence suggests that dysregulation of long non-coding RNAs (lncRNAs) are associated with many neurological diseases (*Gao et al., 2022*), including IS (*Wu et al., 2022*). This involves lncRNAs acting as competitive endogenous RNAs (ceRNAs) to sponge microRNAs (miRNAs) (*Li et al., 2021a*). SNHG10 has been reported to be a therapeutic target in various cancers, such as liver cancer (*Lan et al., 2019*) and colorectal cancer (*Zhang et al., 2021*), and knockdown of its expression significantly ameliorated damage and inflammation in SH-SY5Y cells induced by 1-methyl-4-phenylpyridine ion (*Jin et al., 2020*). However, the mechanism of action of SNHG10 in IS remains unclear.By attaching to the 3′ untranslated region (UTR) of target mRNAs, miRNAs control the expression of genes, thereby affecting the pathological state of IS (*Eyileten et al., 2021*). Among them, miR-665 is an important regulator of many diseases, such as bladder cancer (*Wang et al., 2021*) and heart failure (*Fan et al., 2020*). Studies have shown that miR-665 is also involved in IS progression (*Ouyang et al., 2022*) and that its upregulation can alleviate apoptosis and inflammation caused by oxygen-glucose deprivation and reoxygenation (OGD/R) (*Zhang et al., 2020*). RASSF5 is a member of the Ras-association domain family (RASSF), and its depletion reduces the number of cells without axons, which are involved in neuronal polarization (*Nakamura et al., 2013*). Studies have shown that miR-27a-3p ameliorates disc degeneration by targeting RASSF5, and RASSF5 promotes apoptosis and inhibits the proliferation of nucleus pulposus cells (*Yuan et al., 2022*). However, the role and mechanism of action of RASSF5 in IS, and its association with miR-665, remain unclear.

Dihydromyricetin (DHM), with known antioxidant and anti-inflammatory properties, may interact with this axis, potentially modulating the lncRNA/miRNA pathways that converge on nuclear factor-kappa B (NF-κB) to reduce IS-induced neuroinflammation and cell death. NF-κB pathway is a key mediator of inflammatory responses, oxidative stress, and cell survival (*Guo et al., 2021*), which is rapidly activated after ischemic injury and persists during reperfusion, potentially exacerbating ischemic injury (*Xian et al., 2021*). During this process, increased neuroinflammation and ROS accumulation under OGD/R stimulation promote neuronal cell death (*Chen et al., 2023*; *Xiao et al., 2021*). This provides theoretical support for our study using OGD/R to stimulate SH-SY5Y cells to establish an *in vitro* model simulating the *in vivo* IS environment, which is a mature *in vitro* IS model.

DHM is a flavonoid compound isolated from Ampelopsis grossedentata (*Watanabe et al., 2022*) that has antioxidant, anti-inflammatory, and anti-apoptotic properties, and its potential as a neuroprotective agent has been demonstrated (*Wang et al., 2022b*). Study have shown that DHM ameliorates IS by reducing apoptosis and oxidative stress stimulated by OGD/R (*Tao et al., 2022*). However, the mechanisms of action of DHM in the treatment of IS have not yet been fully elucidated.

This study aims to elucidate the role of SNHG10 in the pathology of ischemic stroke, specifically its regulatory effects on miR-665 and related inflammatory and oxidative stress pathways, providing insights into potential therapeutic strategies of DHM for OGD/R-induced injury in SH-SY5Y cells.

## METHODS

### Cell culture

Human neuroblastoma SH-SY5Y cells were purchased from the American Type Culture Collection (Manassas, VA, USA), maintained in DMEM (Gibco, Thermo Fisher Scientific, Waltham, MA, USA) supplemented with 10% fetal bovine serum (Gibco), and incubated at 37 °C.

Cells were incubated in glucose-free DMEM (Gibco) and placed in an anaerobic chamber (Model 10; Thermo Fisher Scientific) filled with a gas mixture of 95% $N_2$ and 5% $CO_2$ for 4 h to induce OGD. The cells were then returned to normal culture conditions (HF100 incubator; Heal force, Shanghai, China) in glucose-containing DMEM and 21% $O_2$ for 24 h to allow reoxygenation. Cells not subjected to OGD/R served as the negative control, which were cultured under standard conditions, with DMEM supplemented with 10% fetal bovine serum at 37 °C and 21% $O_2$. For drug treatment, DHM (Sigma-Aldrich) was dissolved in dimethyl sulfoxide (DMSO; Beyotime, Shanghai, China) and added to the cells at concentrations of 5, 10, 20, 50, and 100 μM 1 h before OGD/R and incubated for 24 h. An equal volume of DMSO was added to the cell culture medium as a negative control (NC), and the final concentration was 0.01% for the assay of toxicity to cells.

### Cell transfection

The SNHG10 overexpression vector was constructed by cloning full-length SNHG10 into a pcDNA3.1 vector (RiboBio, Guangzhou, China), and the empty pcDNA3.1 vector served as the NC. miR-665 mimic, miR-665 inhibitor, their NCs (mimic NC and inhibitor NC, respectively), and three short interfering RNAs (siRNAs) targeting RASSF5 (si-RASSF5) and its NC (si-NC; scrambled) were purchased from Qiagen (Hilden, Germany) (sequences are shown in Table 1). Transfections were performed using Lipofectamine 2000 reagent (Invitrogen, Thermo Fisher Scientific).

### Cell viability assay

The viability of the SH-SY5Y cells was assessed using an MTT assay kit (Solarbio, Beijing, China). Cells ($3\times 10^3$/well) were inoculated into 96-well plates overnight., After treatment with different concentrations of DHM and/or exposure to OGD/R to simulate ischemic injury, a 20 μL aliquot of MTT solution (5 mg/mL) was added to each well. The cells were then incubated at 37 °C for 4 h to allow the formation of formazan crystals. Following incubation, the MTT solution was carefully removed, and 150 μL of DMSO was added to dissolve the crystals. The plate was shaken gently to ensure complete dissolution. Absorbance was measured at 570 nm using a microplate reader (Multiskan FC Microplate Reader; Thermo Fisher Scientific), with cell viability expressed as a percentage relative to untreated control cells.

### Nucleocytoplasmic separation and RT-qPCR

RNA was isolated from cells using a Cytoplasmic and Nuclear RNA Purification Kit (Norgen Biotek, Ontario, Canada). Total RNA was extracted using TRIzol reagent (Invitrogen), and RNA purity was assessed by measuring the A260/A280 ratio using a
**Table 1  Sequences of the 3 siRNAs targeting RASSF5 and miR-665 mimic or inhibitor.**

| Sequences 5′-3′ | |
|---|---|
| si-RASSF5-1 sense | GGCAUAUAGCUAUAUAUAAAG |
| si-RASSF5-1 antisense | UUAUAUAUAGCUAUAUGCCUU |
| si-RASSF5-2 sense | GUUACAAAUUUGAAUUUAAUG |
| si-RASSF5-2 antisense | UUAAAUUCAAAUUUGUAACUU |
| si-RASSF5-3 sense | AGCAGAAGAUCGACAGCUACA |
| si-RASSF5-3 antisense | UAGCUGUCGAUCUUCUGCUUG |
| si-NC sense | GAACCGGAAACAGGAACCAUU |
| si-NC antisense | UGGUUCCUGUUUCCGGUUCUU |
| Mimic NC sense | GAGCGUGCCCUGAGCCACAG |
| Mimic NC antisense | CUGUGGCUCAGGGCACGCUC |
| MiR-665 mimic sense | ACCAGGAGGCUGAGGCCCCU |
| miR-665 mimic antisense | AGGGGCCUCAGCCUCCUGGU |
| Inhibitor NC | CAGUACUUUUGUGUAGUACAA |
| miR-665 inhibitor | AGGGGCCUCAGCCUCCUGGU |

spectrophotometer (Thermo Fisher Scientific), with values between 1.8 and 2.0 indicating high purity. Complementary DNA (1 μg) was synthesized using the PrimeScript RT Reagent Kit (Takara Bio, Shiga, Japan). qPCR was performed using SYBR Green PCR Master Mix (Applied Biosystems, Foster City, CA, USA) on a StepOne Real-Time PCR System (Applied Biosystems). Thermal cycling conditions included an initial denaturation at 95 °C for 30 s, 40 cycles of denaturation at 95 °C for 15 s, annealing at 60 °C for 30 s, and extension at 72 °C for 30 s. RNA levels of lncRNA SNHG10, RASSF5, and miR-665 were calculated using the $2^{-\Delta\Delta Ct}$ method with GAPDH or U6 as internal references for normalization (*Singaravelan, Sivaperuman & Calambur, 2022*) (primer sequences are shown in Table 2). All experiments were repeated three times.

## ROS analysis

The ROS levels were measured using a spectrophotometer-based reactive oxygen species (ROS) assay kit (S0033, Beyotime). This assay utilizes the DCFH-DA fluorescent probe (Beyotime), which penetrates the cell membrane and is hydrolyzed intracellularly by esterases to form non-fluorescent DCFH. Cells were incubated with DCFH-DA (10 μM final concentration) in serum-free medium for 30 min at 37 °C in the dark. The oxidation of DCFH-DA by intracellular ROS produced the fluorescent compound DCF, and fluorescence intensity was quantified to indicate ROS levels. The fluorescence intensity of DCF, which correlates with ROS levels, was measured at an excitation wavelength of 488 nm and an emission wavelength of 525 nm, providing quantitative analysis of intracellular ROS production.

## TUNEL analysis

The TUNEL assay was performed using the In Situ Cell Death Detection Kit (Beyotime). SH-SY5Y cells were permeabilized with 0.1% Triton X-100 in 0.1% sodium citrate (Beyotime) for 8 min at 21 °C. After washing with PBS, sections were incubated with

**Table 2  Primer sequences used for RT-qPCR analysis.**

| Primers | Forward primer 5′-3′ | Reverse primer 5′-3′ |
| --- | --- | --- |
| miR-665 | ACACTCCAGCTGGGACCAGGAGGCTGAG | CTCAACTGGTGTCGTGGA |
| miR-2278 | ACACTCCAGCTGGGGAGAGCAGTGTGTGTT | CTCAACTGGTGTCGTGGA |
| miR-6884-5p | ACACTCCAGCTGGGAGAGGCTGAGAAGGTG | CTCAACTGGTGTCGTGGA |
| miR-485-5p | ACACTCCAGCTGGGAGAGGCTGGCCGTGAT | CTCAACTGGTGTCGTGGA |
| miR-3196 | ACACTCCAGCTGGGCGGGGCGGCAGG | CTCAACTGGTGTCGTGGA |
| miR-3180 | ACACTCCAGCTGGGTGGGGCGGAGCTT | CTCAACTGGTGTCGTGGA |
| miR-6816-5p | ACACTCCAGCTGGGTGGGGCGGGGCAGGT | CTCAACTGGTGTCGTGGA |
| miR-3180-3p | ACACTCCAGCTGGGTGGGGCGGAGCTTCCG | CTCAACTGGTGTCGTGGA |
| U6 | CTCGCTTCGGCAGCACA | AACGCTTCACGAATTTGCGT |
| SNHG10 | AAGCTTGGACCCATCGTGAG | GCCTGATGAGGCTTGCTTTG |
| RASSF5 | TGCGGAGCATCTTCGAGC | TGACAGGTGAATTTACAGTTAGTGC |
| NTMT1 | GAATGCCCACTCCAACACCT | CAACCGCGACTCTCCTGG |
| LIX1L | GGGCTATGGCCGAGTGAAT | CTGGAAACTCCCAAAGCAGC |
| SYNGR2 | TGCTCTTCTCAGGGTGTGCT | CGGAGTGGGGTCAACGTAAT |
| COPS7B | CAAATTCACAAACCCGCCCG | TAGCTCCTTCCGCAAGCATGA |
| ZNF740 | CGCCTGGCAGGTTTCTGA | CCCATCAGTCCTGTTGCCAT |
| GAPDH | GCTCATTTGCAGGGGGGAG | GTTGGTGGTGCAGGAGGCA |

TUNEL reaction mixture at 37 °C for 60 min. Sections were then washed three times with PBS and counterstained with 4′,6-diamidino-2-phenylindole (DAPI; Invitrogen) for 5 min to visualize cell nuclei.

## Enzyme-linked immunosorbent assay (ELISA)

Levels of inflammatory cytokines (TNF-α, IL-6, IL-1β, and IL-18) and oxidative stress markers (MDA and SOD) in the cell culture supernatant were measured using ELISA kits (Solarbio). SH-SY5Y cells were treated under various experimental conditions, and supernatants were collected after centrifugation at 1,000× g (Centrifuge 5810R; Eppendorf, Hamburg, Germany) for 10 min to remove any cell debris. For each target, 100 μL of the cell culture supernatant was added to individual wells of 96-well ELISA plates pre-coated with specific antibodies for each marker. Plates were incubated at 37 °C for 1 h. After incubation, wells were washed five times with the provided wash buffer to remove unbound components. Subsequently, 100 μL of horseradish peroxidase (HRP)-conjugated secondary antibody was added to each well, followed by a 30-minute incubation at 37 °C. Following further washing, 100 μL of TMB (3,3′,5,5′-tetramethylbenzidine, Solarbio) substrate solution was added to develop color, and plates were incubated in the dark for 15–30 min at room temperature. The reaction was then stopped by adding 50 μL of stop solution (Solarbio), changing the color from blue to yellow. Absorbance was measured at 450 nm using a microplate reader (Multiskan FC Microplate Reader; Thermo Fisher Scientific).

## Bioinformatic analysis

Target prediction was performed using the starBase database for lncRNA SNHG10, and starBase and TargetScan databases for miR-665. The likelihood of the outcome predicted
by the TargetScan database is extremely high when the context++ score is ≥90. The interaction between miR-665 and predicted targets was further verified using a luciferase reporter assay.

## Luciferase assay

The psiCHECK-2 dual luciferase reporter vector (Promega, Madison, WI, USA) was cloned including SNHG10 and 3′-UTR segments of RASSF5 that included miR-665 binding sites. The reporter vector and the miR-665 mimic were co-transfected into SH-SY5Y cells using Lipofectamine 2000 for 48 h. A dual-luciferase reporter assay system (Promega) was used to quantify luciferase activity, according to the manufacturer's instructions. Firefly luciferase data were normalized to the Renilla luciferase activity.

## RNA pulldown

Biotinylated lncRNA SNHG10 (bio-SNHG10), a mutant version of bio-SNHG10 (bio-mut), and a biotin-labeled non-specific control RNA (bio-NC) were synthesized (RiboBio, Guangzhou, China). SH-SY5Y cells were transfected with 50 nM of each biotinylated RNA construct using Lipofectamine 2000 (Invitrogen), and cells were harvested 48 h post-transfection. After cell lysis in RIP lysis buffer supplemented with protease and RNase inhibitors, cell lysates were incubated with streptavidin-coated magnetic beads (Dynabeads M-280 Streptavidin; Invitrogen) at 4 °C for 4 h to allow binding of the biotin-labeled RNA to the beads. The bead-bound complexes were then washed five times with cold wash buffer to remove non-specifically bound proteins and RNAs. After washing, RNA was extracted from the complexes using TRIzol reagent (Invitrogen). The co-precipitated RNAs were then quantified by RT-qPCR (*Hu et al., 2023*).

## Western blot analysis

Using sodium dodecyl sulfate-polyacrylamide gel electrophoresis, protein samples were prepared in radioimmunoprecipitation assay lysis buffer (Solarbio) and transferred to a polyvinylidene difluoride membrane (Millipore, Burlington, MA, USA). The membranes were blocked with 5% BSA and incubated with RASSF5 (1:300, ab204117), NF-κB (1:1000, ab32536), phosphor (p) NF-κB (1:1000, ab76302), and GAPDH (1:2500, ab9485) primary antibodies, followed by HRP-conjugated secondary antibodies (1:3000, ab6721). All antibodies were purchased from Abcam (Cambridge, UK). Protein bands were visualized using ECL detection reagent (Thermo Fisher Scientific).

## Statistical analysis

All experiments were repeated three times, data are presented as mean ± standard deviation. Statistical analyses were performed using SPSS statistical analysis software (version 22.0, IBM Corp., NY, USA), and significance was determined by one-way analysis of variance (ANOVA) with Tukey's post hoc test, statistical significance defined as $p < 0.05$. Graphical representations, including bar charts, were created using GraphPad Prism 8.0 (GraphPad Software, Inc., La Jolla, CA, USA).

## RESULTS

### The effect of DHM on SH-SY5Y viability

Since 50 and 100 µM DHM (chemical structure in Fig. 1) caused a slight inhibition of cell viability ($p < 0.05$), we chose 5, 10, and 20 µM as safe concentrations of DHM for subsequent experiments (Fig. 2A). Based on the observed concentration-dependent effects on cell viability, we further investigated the influence of DHM on other cellular processes affected by OGD/R. Our data revealed that DHM ameliorated OGD/R-induced cell viability (Fig. 2B), apoptosis (Fig. 2C). neuroinflammation (increased TNF-α, IL-6, IL-1β, and IL-18) (Figs. 3A–3D), and oxidative stress (increased ROS and MDA, decreased SOD) (Figs. 3E–3G) in a dose-dependent manner. Interestingly, overexpression of lncRNA SNHG10 in the OGD/R response was inhibited by DHM ($p < 0.05$) (Fig. 4A), suggesting the potential involvement of SNHG10 in the protective mechanism of DHM. Furthermore, cytoplasmic-nuclear separation analysis revealed abundant SNHG10 expression in the cytoplasm, similar to that of the GAPDH positive control, but in contrast to U6 (Fig. 4B).

### The functionality of SH-SY5Y cells was improved by DHM *via* SNHG10

To further investigate the role of SNHG10, an SNHG10 overexpression vector was constructed and transfected into SH-SY5Y cells. The efficacy of this overexpression was confirmed through RT-qPCR analysis ($p < 0.05$), as shown in Fig. 5A. Upon SNHG10 overexpression, we noted a partial reversal of the DHM-induced protective effects in the cells subjected to OGD/R conditions. Specifically, SNHG10 overexpression led to a reduction in cell viability (Fig. 5B), an increase in apoptosis levels (Fig. 5C), and a significant increase in markers of neuroinflammation, including elevated levels of TNF-α, IL-6, IL-1β, and IL-18 (Figs. 6A–6D). Additionally, oxidative stress markers were adversely affected, with an increase in ROS and MDA levels, coupled with a decrease in SOD activity (Figs. 3E–3G). These observations suggest that SNHG10 may act as a regulatory factor that modulates the protective effects of DHM under OGD/R-induced stress, highlighting its role in influencing cell viability, inflammation, and oxidative stress responses.

### SNHG10 targets binding to miR-665

According to the starBase database analysis, SNHG10 has eight predicted downstream miRNAs: miR-2278, miR-6884-5p, miR-485-5p, miR-3196, miR-3180, miR-6816-5p, miR-3180-3p, and miR-665. Among them, miR-665 had the lowest expression in the OGD/R group, displayed a dose-dependent increase following DHM treatment ($p < 0.05$) (Fig. 7), and was suppressed by SNHG10 overexpression ($p < 0.05$) (Fig. 8A). MiR-485-5p, which was also downregulated in OGD/R but partially counteracted by DHM, was not significantly altered by SNHG10 overexpression ($p < 0.05$) (Fig. 8B). Therefore, we selected miR-665 for further mechanistic studies. After validating the efficacy of the miR-665 mimic/inhibitor (Fig. 8C), we found no significant change in the expression of SNHG10 ($p > 0.05$) (Fig. 8D). Using a dual luciferase assay, it was observed that the luciferase activity of WT-SNHG10 co-transfected with the miR-665 mimic was lower than that of mut-SNHG10 ($p < 0.05$). However, there was no significant difference in luciferase activity between WT-SNHG10 and mut-SNHG10 cells co-transfected with the miR-665 inhibitor

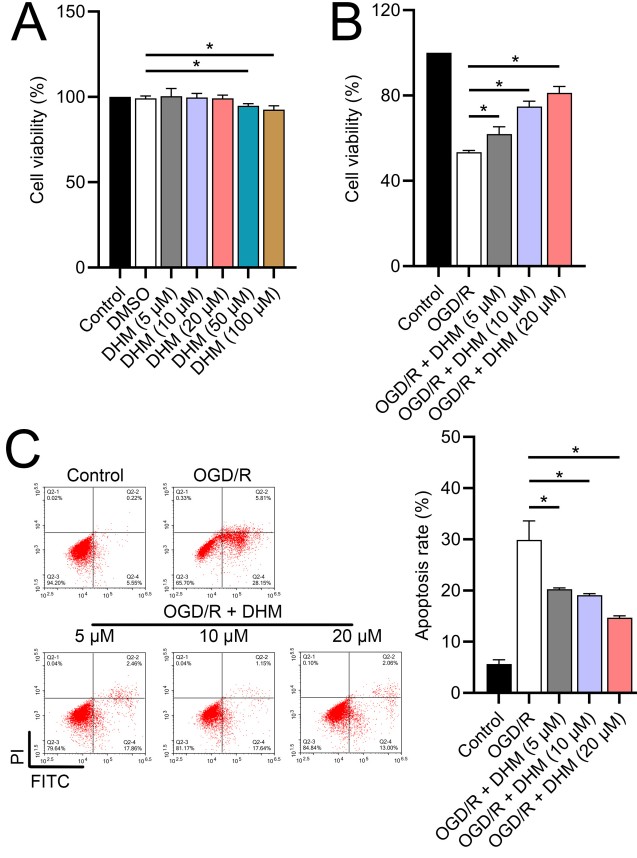

**Figure 1  Chemical structure of DHM, a flavonoid compound investigated for its neuroprotective effects in IS models.**

**Figure 2  Effects of DHM on SH-SY5Y cell viability and apoptosis during OGD/R treatment.** (A) MTT analysis of the effect of DHM on viability in SH-SY5Y cells. (B) MTT assay showing the effects of DHM on SH-SY5Y cell viability during OGD/R treatment. (C) Flow cytometry quantifying apoptosis rates following DHM treatment, indicating decreased apoptosis in OGD/R-treated cells. *$P < 0.05$.

(Fig. 8E). RNA pulldown results showed that miR-665 was enriched in the bio-SNHG10 group compared to the bio-NC and bio-mut groups (Fig. 8F), confirming that miR-665 is a direct target of SNHG10.

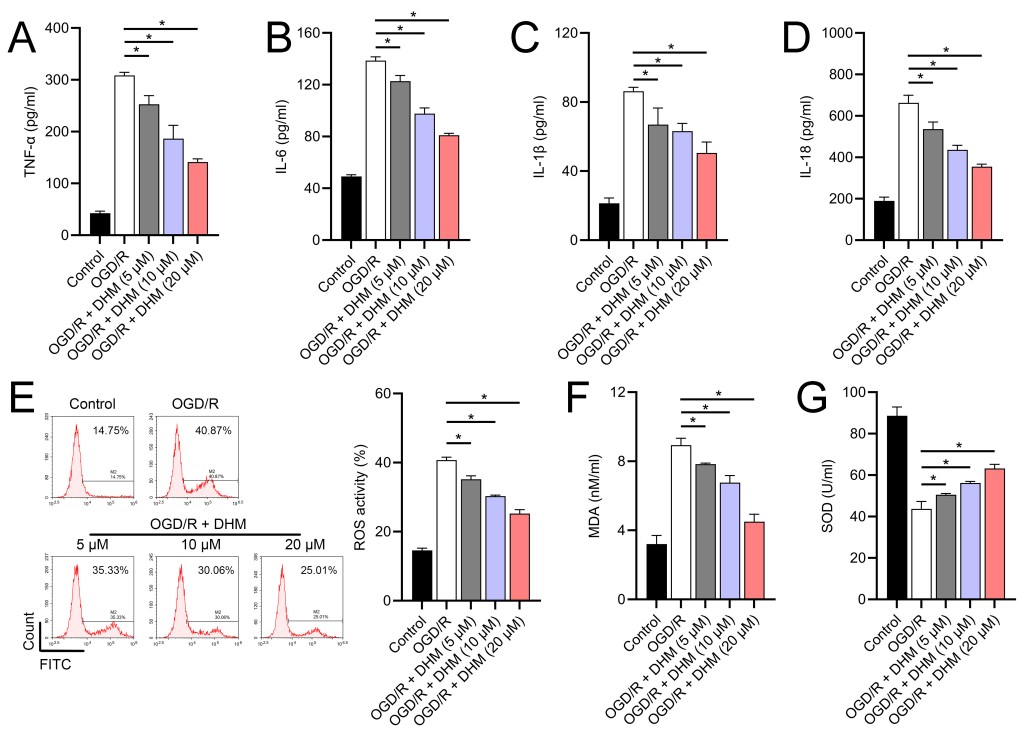

**Figure 3** **Effects of DHM on SH-SY5Y cell neuroinflammatory markers, and oxidative stress during OGD/R treatment.** (A–D) ELISA showing the effects of DHM on neuroinflammatory markers TNF-α, IL-6, IL-1β, IL-18. (E) Flow cytometry analysis of ROS levels after DHM treatment. (F, G) ELISA showing the influence of DHM on oxidative stress markers ROS and MDA and SOD activity. $\star P < 0.05$.

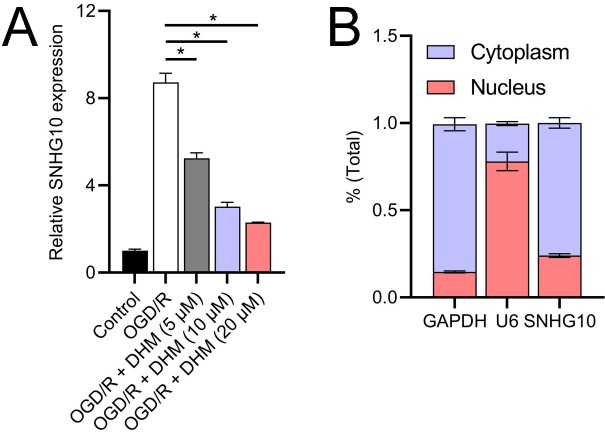

**Figure 4** **Effects of DHM on SNHG10 expression during OGD/R treatment.** (A) RT-qPCR analysis showing the expression of lncRNA SNHG10 inhibited by DHM during OGD/R response. (B) Nucleoplasmic isolation assay to analyze the localization of lncRNA SNHG10 in SH-SY5Y cells. $\star P < 0.05$.

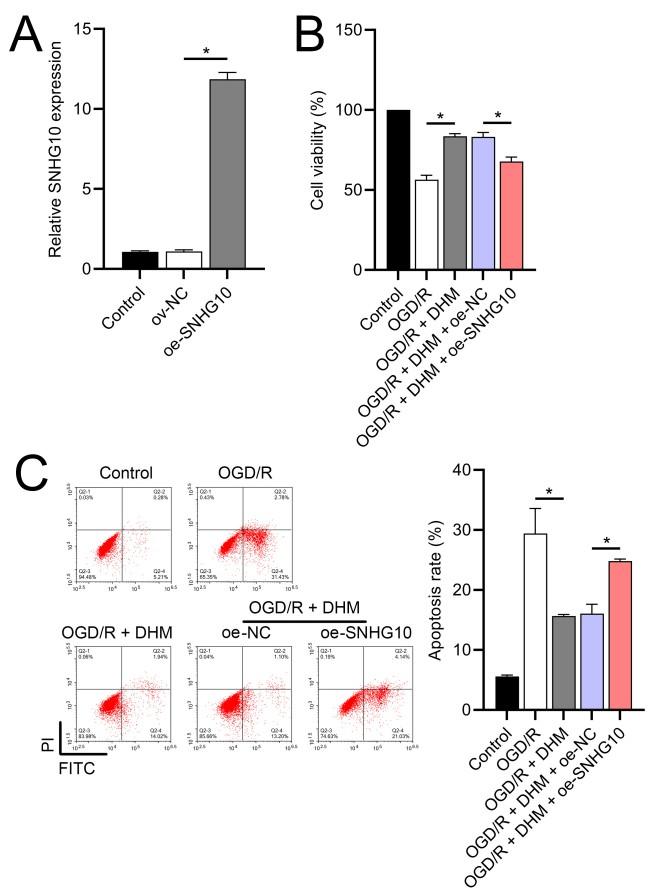

**Figure 5  Effects of SNHG10 overexpression on SH-SY5Y cells.** (A) RT-qPCR confirming the overexpression of SNHG10 in SH-SY5Y cells. (B) MTT assay showing the effect of DHM on cell viability after SNHG10 transfection during OGD/R treatment. (C) Flow cytometry analysis showing the apoptosis rate after SNHG10 transfection. $*P < 0.05$.

## RASSF5 is a direct target of miR-665

Through a combined analysis of the TargetScan and starBase databases, we identified 60 genes as potential targets of miR-665, with these genes achieving a context++ score of $\geq 90$ in the TargetScan database (Fig. 9A). By excluding genes that were previously demonstrated to play significant roles in IS or SH-SY5Y cells, we identified six genes whose roles and mechanisms in IS or SH-SY5Y cells remained unverified. These genes included RASSF5, NTMT1, LIX1L, SYNGR2, COPS7B, and ZNF740. We then performed RT-qPCR analysis of these six genes, similar to the miRNA screening approach. The results showed that the expression of RASSF5 and LIX1L was increased by OGD/R treatment and was suppressed by DHM (Figs. 9B–9G). However, RASSF5 expression increased with SNHG10 overexpression ($p < 0.05$) (Fig. 10A), whereas LIX1L expression did not significantly change ($p > 0.05$) (Fig. 10B). Therefore, we selected RASSF5 as a potential target of miR-665 for further experiments. RASSF5 expression and protein levels were negatively regulated by miR-665 (Figs. 10C, 10D). TargetScan database predicted the binding site between RASSF5 and

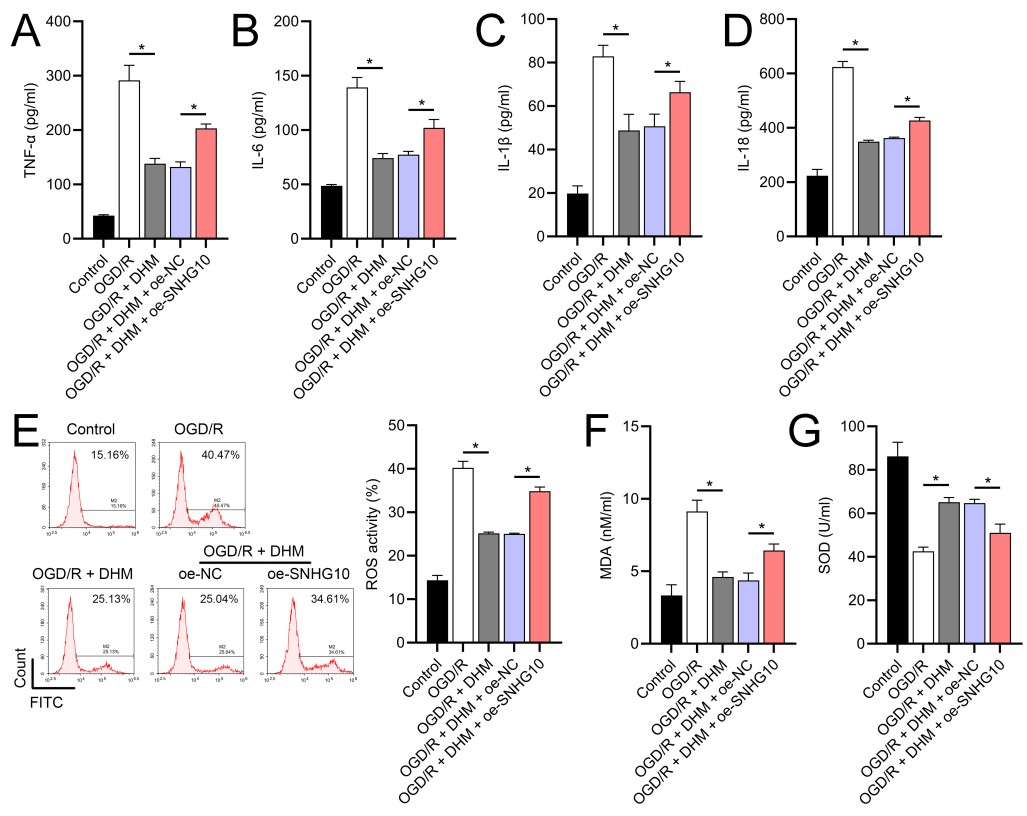

**Figure 6  Effects of oe-SNHG10 on inflammation and oxidative stress in SH-SY5Y cells under DHM.**
(A–D) ELISA showing the effects of SNHG10 overexpression on neuroinflammatory markers TNF-α, IL-6, IL-1β, IL-18. (E) Flow cytometry analysis showing ROS levels after SNHG10 transfection. (F, G) ELISA showing the effects of SNHG10 overexpression on oxidative stress markers MDA and SOD activity. *$P$ < 0.05.

miR-665 (Figs. 10E). Luciferase activity of WT-RASSF5 co-transfected with the miR-665 mimic was lower than that of mut-RASSF5 ($p < 0.05$). However, there was no significant difference in luciferase activity between WT-RASSF5 and mut-RASSF5 cells co-transfected with the miR-665 inhibitor (Fig. 10F). Therefore, RASSF5 is a direct target of miR-665. The mRNA levels of RASSF5 were downregulated in the si-RASSF5-transfected group compared to those in the si-NC group (Fig. 10G), indicating the successful synthesis of si-RASSF5, with si-RASSF5-1 having the best inhibitory effect.

## DHM improves SH-SY5Y cells *via* SNHG10/miR-665/RASSF5 axis

To investigate the interaction between DHM and the SNHG10/miR-665/RASSF5 axis, we co-transfected SNHG10 cells with miR-665 mimic/si-RASSF5. The results showed that DHM ameliorated the OGD/R-mediated decrease in SH-SY5Y cell viability (Fig. 11A), apoptosis (Fig. 11B), neuroinflammation (TNF-α, IL-6, IL-1β, and IL-18) (Figs. 12A–12D), oxidative stress (increased ROS and MDA, decreased SOD) (Figs. 12E–12G), and increased RASSF5 expression (Fig. 13A). Intervention with miR-665 promoted the effect of DHM, and the counteracting effect of SNHG10 on miR-665 was inhibited by si-RASSF5. In

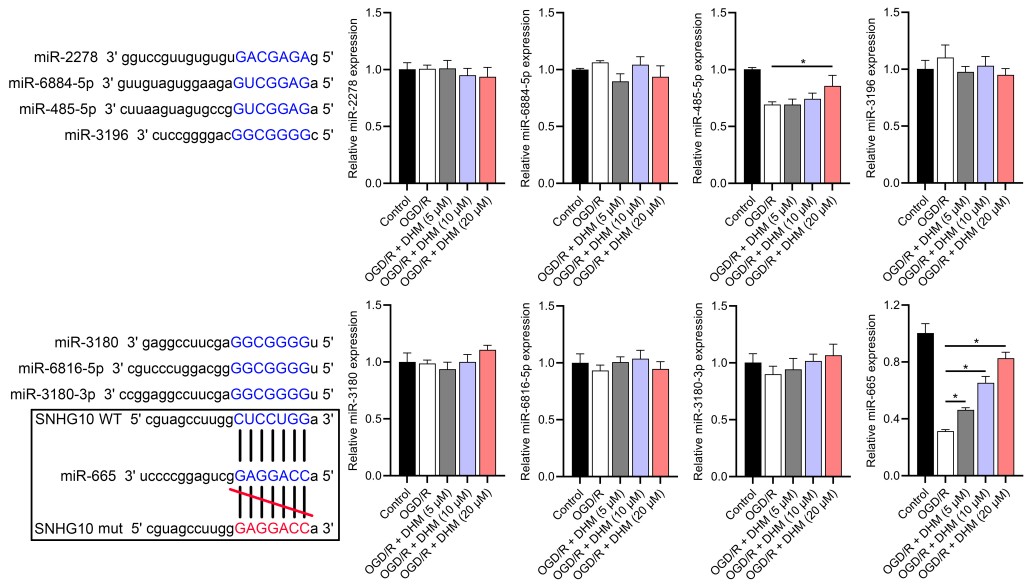

**Figure 7  Effects of SNHG10 overexpression on SH-SY5Y cells.** RT-qPCR analysis of the expression changes of eight putative downstream miRNAs of SNHG10 in response to OGD/R and DHM treatment. These miRNAs were miR-2278, miR-6884-5p, miR-485-5p, miR-3196, miR-3180, miR-6816-5p, miR-3180-3p, and miR-665. *$P < 0.05$.

addition, DHM alleviated OGD/R-induced elevated RASSF5 protein levels and NF-κB phosphorylation, and the addition of miR-665 mimic enhanced the effect of DHM. si-RASSF5 attenuated the effect of SNHG10 on the miR-665 mimic (Fig. 13B). Thus, DHM protected SH-SY5Y cells by reducing the transcription and translation of RASSF5 and inactivating the NF-κB signaling pathway through the SNHG10/miR-665 axis.

## DISCUSSION

This study revealed the significant role and underlying molecular mechanisms of SNHG10 in the neuroprotective effects of DHM against ischemic injury, involving the regulation of inflammation and oxidative stress, which are key factors influencing IS (*Wang et al., 2022a*). Our findings reveal a novel regulatory axis, SNHG10/miR-665/RASSF5, which modulates neuroinflammation and oxidative stress in ischemic stroke, suggesting this axis as a promising therapeutic target.

This study aimed to understand the effects of DHM on SH-SY5Y cells exposed to OGD/R. Notably, DHM enhanced cell viability and reduced apoptosis in a dose-dependent manner, confirming previous studies that emphasized the anti-apoptotic effects of DHM (*Xie et al., 2022*). In addition, DHM suppressed inflammation and oxidative stress. This observation was supported by the decreased levels of inflammatory factors and oxidative stress markers and the increase in antioxidant enzyme SOD. These results suggest that DHM may exert neuroprotective effects by attenuating OGD/R-induced cellular injury, which is consistent with its known antioxidant and anti-inflammatory properties (*Li et al., 2021b*). Although previous studies have investigated the role of SNHG10 in various

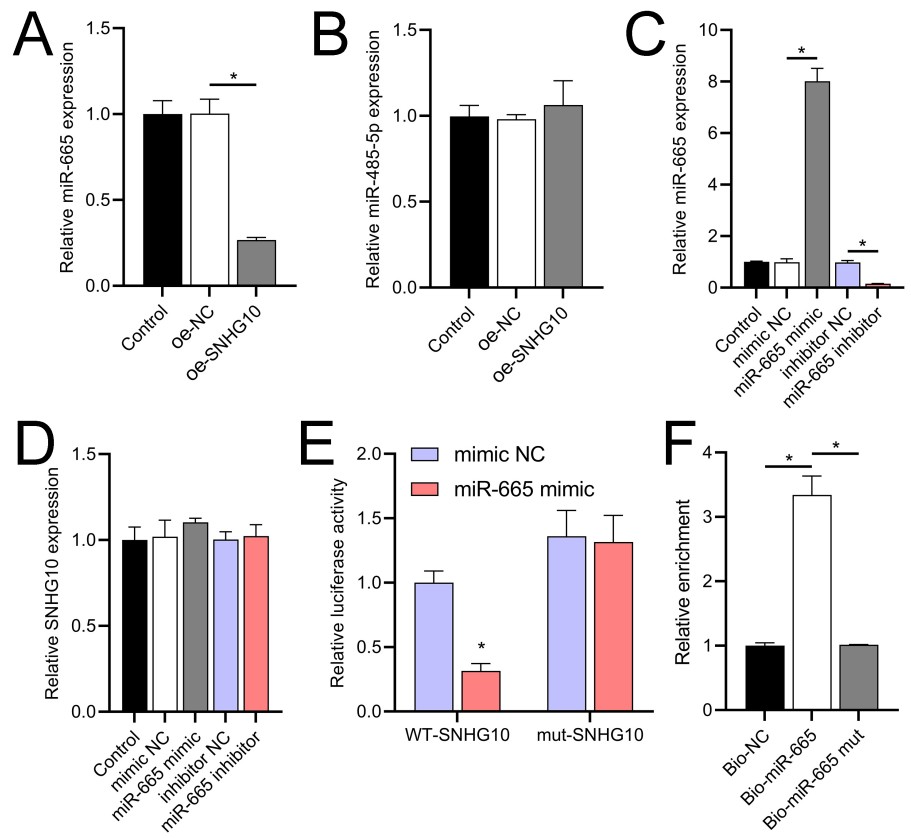

**Figure 8** **Effects of SNHG10 overexpression on SH-SY5Y cells.** (A, B) RT-qPCR analysis of the effect of SNHG10 overexpression on miR-665 (A) and miR-485-5p (B) expression. (C) RT-qPCR to validate the efficacy of miR-665 mimic/inhibitor. (D) RT-qPCR analysis of the effect of miR-665 mimic/inhibitor on SNHG10 expression. (E) Dual-luciferase reporter assay confirming miR-665 as a direct target of lncRNA SNHG10. (F) RNA pulldown analysis of miR-665 enrichment in bio-NC, bio-SNHG10, bio-SNHG10-mut groups. $*P < 0.05$.

diseases (*Aini et al., 2022*; *Ge et al., 2022*), its function and mechanism of action in IS remain unclear. A key finding of this study was that DHM suppressed OGD/R-induced overexpression of SNHG10, suggesting that SNHG10 may play a role in the protective mechanism of DHM. Increasing the expression of SNHG10 in SH-SY5Y cells completely or partially abolished the protective effects of DHM, suggesting a negative regulatory role of SNHG10 in OGD/R-induced damage. This was consistent with the results of previous studies on SNHG10 promoting inflammation and oxidative stress (*Sun, Song & Li, 2022*). Although the functions of SNHG10 in the central nervous system (CNS) and IS have not been extensively investigated, our study provides new insights into the potential roles of this lncRNA in neuroprotective responses.

Because lncRNAs typically function as ceRNAs to sponge miRNAs and regulate mRNA expression, affecting downstream signaling pathways may alter the course of IS (*Cui et al., 2022*; *Wu et al., 2022*). This includes NF-κB, which is activated under IS conditions (*Xu et al., 2021*). Further studies are required to elucidate the underlying mechanisms.

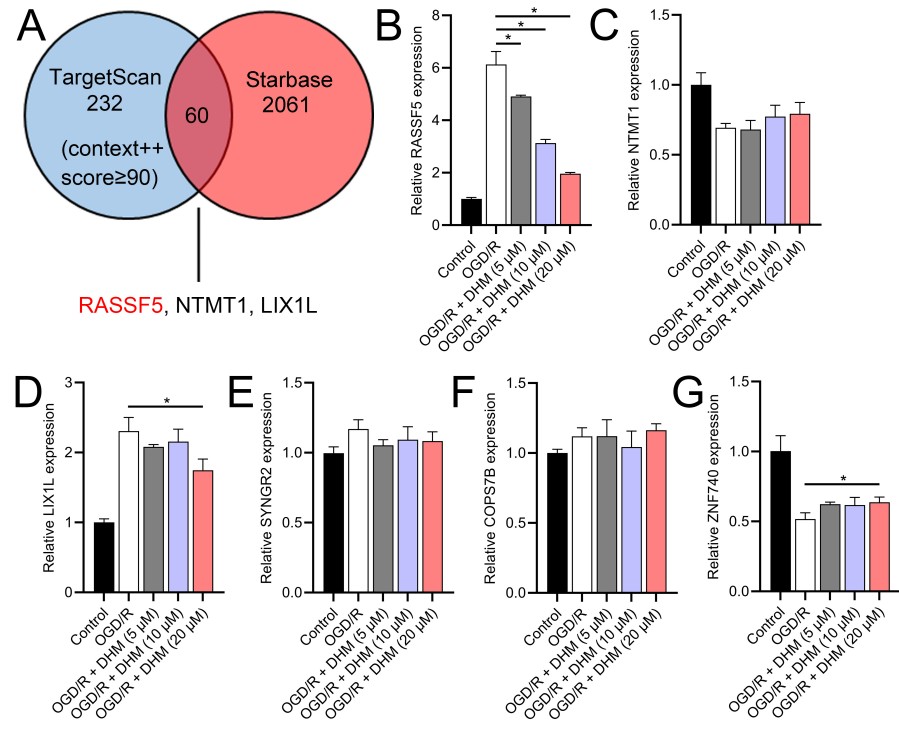

**Figure 9 RASSF5 as a target of miR-665.** (A) Bioinformatics analysis (TargetScan and starBase databases) identified potential downstream target genes of miR-665. (B–G) RT-qPCR analyzed the expression changes of RASSF5, NTMT1, LIX1L, SYNGR2, COPS7B, and ZNF740 in OGD/R-treated SH-SY5Y cells in the presence of DHM. *$P < 0.05$.

Our database analysis identified miR-665 as a potential target of SNHG10, which is a critical regulator of IS progression (*Liu et al., 2022*). The downregulation of miR-665 in OGD/R cells was significantly reversed by DHM treatment in a dose-dependent manner. However, changes in miR-665 expression had no significant effect on SNHG10, further confirming that miR-665 is a downstream target of SNHG10. The binding of miR-665 to SNHG10 was confirmed by dual-luciferase and RNA pulldown experiments, revealing a novel molecular axis in the neuroprotective effects of DHM. In searching for downstream targets of miR-665, we identified RASSF5 as a potential candidate, being an important regulator affecting cell apoptosis and the cell cycle (*Zhou et al., 2014*), which is also related to neuronal polarization (*Nakamura et al., 2013*). However, current research on the mechanism of RASSF5 in IS is lacking. Our results showed that RASSF5 expression was promoted by OGD/R and that SNHG10 was inversely regulated by miR-665, suggesting a potential SNHG10/miR-665/RASSF5 regulatory axis in OGD/R-induced cellular damage. Therefore, this represents a novel regulatory axis. We further verified that RASSF5 is a direct target of miR-665 using luciferase reporter gene detection, providing stronger evidence for the involvement of this axis in ischemic injury. This is also the first time that RASSF5 has been shown to be upregulated under OGD/R stimulation. We later observed that DHM ameliorated OGD/R-induced SH-SY5Y cell injury, inflammation, oxidative stress,

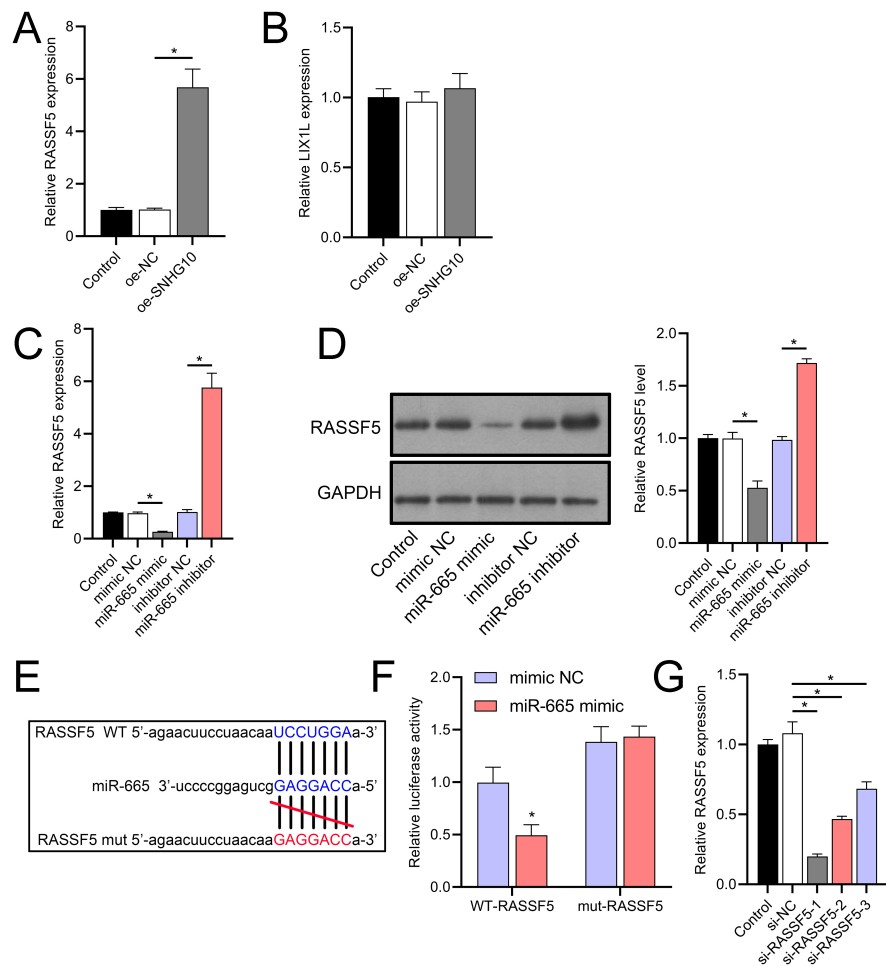

**Figure 10  RASSF5 expression was negatively regulated by miR-665.** (A, B) RT-qPCR analysis showing the effect of SNHG10 overexpression on RASSF5 (A) and LIX1L (B) expression. (C) RT-qPCR analysis of the effect of miR-665 mimic/inhibitor on RASSF5 protein level. (D) Western blot analysis of the effect of miR-665 mimic/inhibitor on RASSF5 expression. (E) Potential binding sites of miR-665 to RASSF5. (F) Dual-luciferase reporter assay confirming RASSF5 as a direct target of miR-665. (G) RT-qPCR to verify the validity of si-RASSF5. $\star P < 0.05$.

and increased NF-κB phosphorylation, with the neutralizing effect of SNHG10 on DHM being offset by miR-665 and si-RASSF5. This suggests that DHM may protect against ischemic injury by modulating the SNHG10/miR-665/RASSF5 axis to inhibit NF-κB signaling, thereby completely or partially neutralizing neuroinflammation and oxidative stress caused by activated NF-κB (*Sarmah et al., 2022*), further validating the interaction of these molecules in IS.

While previous studies have documented DHM's antioxidant, anti-inflammatory, and anti-apoptotic effects in ischemic and neuroprotective contexts, the specific regulatory mechanisms through which DHM acts on lncRNAs and miRNAs, particularly the SNHG10/miR-665/RASSF5 axis, remain unexplored. This study uniquely identifies this pathway as a novel mediator of DHM's protective effects, providing a deeper understanding

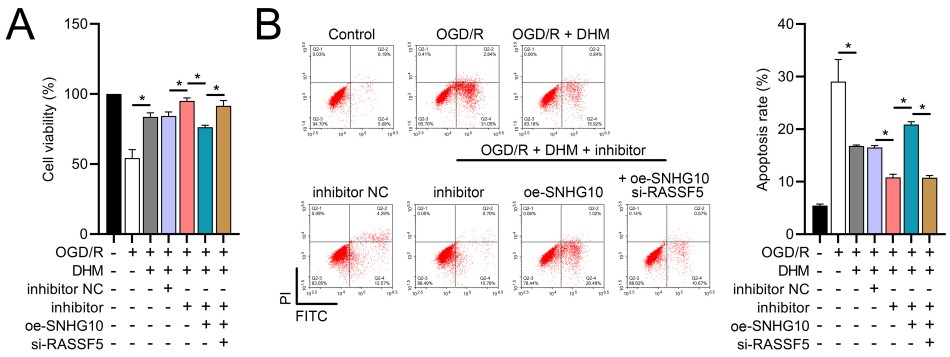

**Figure 11 DHM's modulation of the SNHG10/miR-665/RASSF5 axis improves SH-SY5Y cell growth under ischemic-like OGD/R conditions.** (A) MTT assay evaluating cell viability, showing improved viability in cells treated with DHM, SNHG10, miR-665 mimic, and si-RASSF5.(B) Flow cytometry measuring apoptosis, revealing decreased apoptosis with DHM intervention. $^{\star}P < 0.05$.

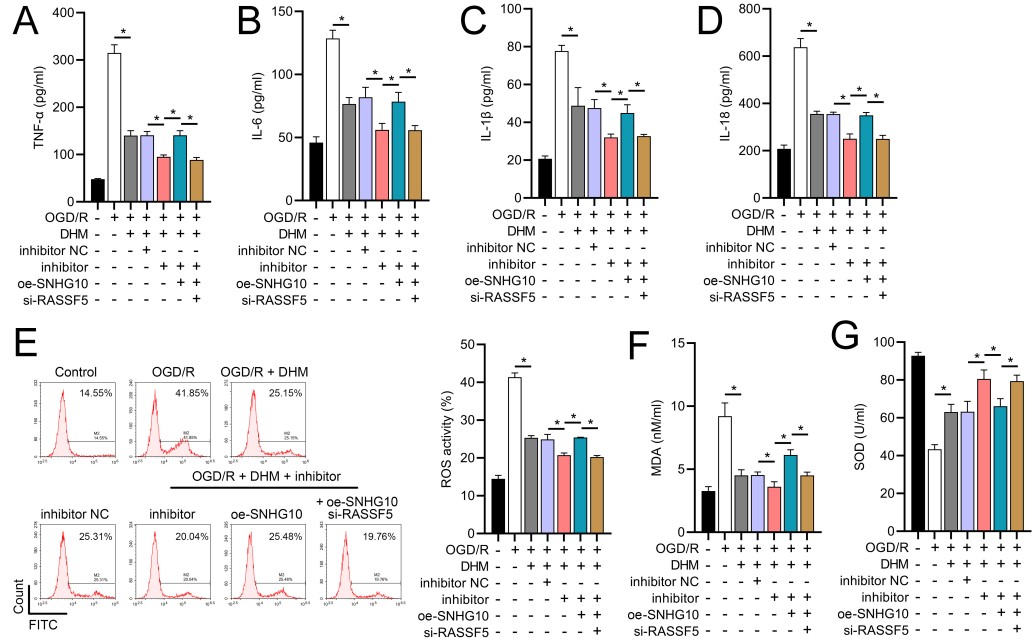

**Figure 12 DHM's modulation of the SNHG10/miR-665/RASSF5 axis improves SH-SY5Y cell inflammation and oxidative stress under ischemic-like OGD/R conditions.** (A–D) ELISA analysis of the interaction between SNHG10, miR-665, RASSF5 on DHM to improve the effect of neuroinflammation markers (TNF-α, IL-6, IL-1β, IL-18) of SH-SY5Y cells stimulated by OGD/R. (E) Flow cytometry analysis of the interaction between SNHG10, miR-665, RASSF5 on DHM to ameliorate the effect of ROS level of SH-SY5Y cells stimulated by OGD/R. (F, G) ELISA analysis of the interaction between SNHG10, miR-665, RASSF5 on DHM to ameliorate the effect of oxidative stress markers MDA and SOD activity of SH-SY5Y cells stimulated by OGD/R.$^{\star}P < 0.05$.

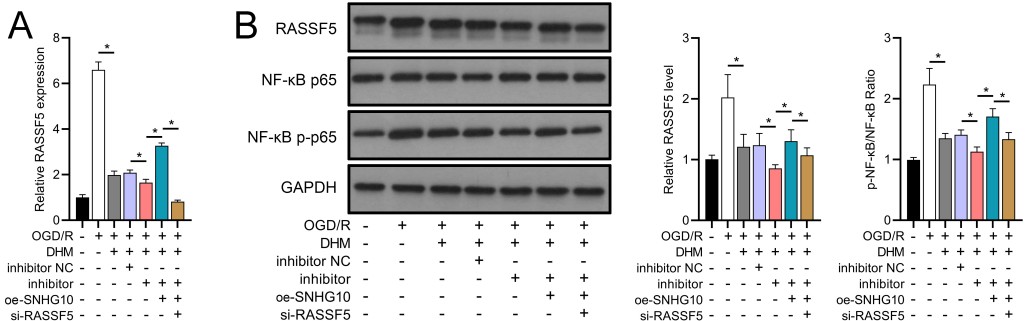

**Figure 13** **DHM's modulation of the SNHG10/miR-665/RASSF5 axis improves SH-SY5Y cell on RASSF5 and NF-$\kappa$B pathway under ischemic-like OGD/R conditions.** (A) RT-qPCR analysis of the interaction between SNHG10, miR-665, RASSF5 on DHM to ameliorate the effect of RASSF5 expression of SH-SY5Y cells stimulated by OGD/R. (B) Western blot analysis of the interaction between SNHG10, miR-665, RASSF5 on DHM to improve the effect of RASSF5, NF-$\kappa$B p65, NF-$\kappa$B p-p65 protein levels of SH-SY5Y cells stimulated by OGD/R. $^{*}P < 0.05$.

of its role in modulating neuroinflammation and oxidative stress. These findings suggest a more targeted therapeutic potential for DHM in ischemic injury by focusing on lncRNA-miRNA interactions, advancing beyond its well-known antioxidative properties.

Our study has several limitations. First, while SH-SY5Y cells offer a convenient *in vitro* model, their neuroblastoma origin may not fully replicate the responses of primary neurons *in vivo*, requires further validation through animal studies and clinical trials to assess its *in vivo* efficacy, safety, and compatibility with existing therapeutic interventions, thereby advancing its clinical application in ischemic stroke treatment. Additionally, transfection efficiency variability could introduce confounding factors, which were minimized by including appropriate controls in each experiment.. Future studies should prioritize *in vivo* experiments to validate the observed *in vitro* effects and better understand the systemic implications of DHM treatment. Second, potential downstream RNAs or signaling pathways of SNHG10 or miR-665 were not the focus of this study. Future studies should delve deeper into these potential pathways to provide a more comprehensive understanding of the underlying molecular interactions. Third, the sample size used in this study, though statistically adequate for preliminary analysis, is limited, and larger sample sizes are recommended to confirm the reproducibility of these results. Fourth, the safety and duration of the effects of DHM have not been fully elucidated. Long-term studies assessing the safety profile and sustained efficacy of DHM are crucial before considering its therapeutic applications. Finally, this study lacked clinical prognostic data for patients with IS using DHM to validate its efficacy. These areas remain future research directions. Future clinical trials and observational studies are required to assess the real-world efficacy and potential side effects of DHM in patients with IS.

In conclusion, this study provides significant insights into the neuroprotective mechanisms of DHM in ischemic stroke, particularly through modulation of the SNHG10/miR-665/RASSF5 axis. We demonstrated that DHM treatment enhances cell viability, reduces apoptosis, and attenuates neuroinflammation and oxidative stress in an

*in vitro* model of ischemic injury. Importantly, these protective effects appear to be mediated by the inhibition of SNHG10 expression and subsequent upregulation of miR-665, which targets RASSF5 and inactivates NF-κB signaling. Our findings suggest that DHM could serve as a promising therapeutic agent for ischemic stroke, providing a novel approach to managing neuroinflammation and oxidative damage. Future studies should focus on further validating these results *in vivo* and exploring DHM's translational potential in clinical applications for stroke therapy.

### Funding

The present study was supported by the Natural Science Foundation of Jiangxi Province (Grant No. 20202BABL206058), the Open Project of Key Laboratory of Prevention and Treatment of Cardiovascular and Cerebrovascular Diseases, Ministry of Education (Grant No. XN201813), the Scientific Research Project of Gannan Medical University (Grant No. TD201804), the Project of Science and Technology of Jiangxi Provincial Health Commission (Grant No. 20195407), and the Project of Science and Technology of the First Affiliated Hospital of Gannan Medical University (Grant No. YJZD202008). The funders had no role in study design, data collection and analysis, decision to publish, or preparation of the manuscript.

### Grant Disclosures

The following grant information was disclosed by the authors:
Natural Science Foundation of Jiangxi Province: 20202BABL206058.
Open Project of Key Laboratory of Prevention and Treatment of Cardiovascular and Cerebrovascular Diseases, Ministry of Education: XN201813.
Scientific Research Project of Gannan Medical University: TD201804.
Project of Science and Technology of Jiangxi Provincial Health Commission: 20195407.
Project of Science and Technology of the First Affiliated Hospital of Gannan Medical University: YJZD202008.

### Competing Interests

The authors declare there are no competing interests.

### Author Contributions

- Qi Zeng conceived and designed the experiments, analyzed the data, authored or reviewed drafts of the article, and approved the final draft.
- Yan Xiao performed the experiments, prepared figures and/or tables, and approved the final draft.
- Xueliang Zeng performed the experiments, prepared figures and/or tables, and approved the final draft.
- Hai Xiao conceived and designed the experiments, analyzed the data, authored or reviewed drafts of the article, and approved the final draft.

## Data Availability

The raw data are available at figshare: Xiao, Hai (2024). Implications of the SNHG10/miR-665/RASSF5/NF-κB pathway in dihydromyricetin-mediated ischemic stroke protection. figshare. Dataset. https://doi.org/10.6084/m9.figshare.26966434.v2.

## Supplemental Information

Supplemental information for this article can be found online at http://dx.doi.org/10.7717/peerj.18754#supplemental-information.

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
