# Peer review of "Implications of the SNHG10/miR-665/RASSF5/NF-κB pathway in dihydromyricetin-mediated ischemic stroke protection"

_PeerJ, doi:10.7717/peerj.18754_

## Round 0.1 · original submission · Major Revisions

The reviewers have raised several important concerns that need to be thoroughly addressed before the manuscript can be considered for publication. While they find your work potentially interesting, major revisions are required to improve the scientific rigor and clarity of presentation. Please provide point-by-point responses to all reviewer comments and revise your manuscript accordingly. The revised version will be reconsidered after adequate revision.

·

Basic reporting

The graphics in this are understandable, however I do not agree with the statistical program used.

Experimental design

Please, explain the design better, if possible make a graphic drawing.
Please, describe cell viability assay in detail.
Please, describe ROS analysis in detail. (probe concentration, ml, etc)
Please, the methodologies in the article should be better detailed.

Pag 94: Gibco …..? (Thermo Fisher Scientific, Waltham, MA, USA ?)
Pag 96-97: (Thermo Fisher Scientific…..) Waltham,MA?
Pag 100: what are the normal conditions? (Sigma-Aldrich….)

Validity of the findings

The research is of great scientific interest, however the methods and results must be more descriptive.

Reviewer 2 ·

Basic reporting

The research focus of this manuscript is clearly defined, exploring the mechanism of the SNHG10/miR-665/RASSF5/NF-κB pathway in the protective effects of dihydromyricetin (DHM) against ischemic stroke. Through in vitro experiments, the study provides an in-depth analysis of DHM’s protective role in ischemic injury via the regulation of non-coding RNAs (lncRNA and miRNA), offering a certain degree of novelty and scientific value, particularly in revealing novel molecular mechanisms. However, the manuscript also presents certain limitations that hinder its potential to further improve in quality and impact.
1.While the abstract summarizes the results, it would benefit from more quantitative data to underscore the significance of findings. Including key metrics such as fold changes or statistical significance would strengthen the impact.
2.The introduction discusses ischemic stroke and related biological processes but could be streamlined to focus more on the specific knowledge gaps addressed by the study, particularly in the context of SNHG10's role in ischemic injury.
3.The role of DHM in protecting against ischemic injury is mentioned but not adequately emphasized in terms of novelty. Highlight how this study differs from prior work on DHM’s protective mechanisms.

Experimental design

4.The choice of SH-SY5Y cells for the OGD/R model should be more clearly justified. Why is this cell line appropriate for studying ischemic stroke mechanisms, and what are its limitations?
5.Certain experimental procedures, such as the specific conditions for OGD/R treatment and cell viability assays, lack sufficient detail. For reproducibility, all experimental conditions, including oxygen concentrations and durations, should be more clearly described.
6.It is unclear whether appropriate negative and positive controls were consistently included for all experiments. For example, controls for transfection efficiency and DHM treatment should be mentioned explicitly.

Validity of the findings

7.The manuscript briefly mentions transfection with SNHG10 overexpression vectors and miR-665 mimics. However, there is no mention of transfection efficiency validation. This should be included to confirm the reliability of the knockdown or overexpression.
8.Each figure should include sufficient detail in the legend to be fully understood independently of the main text.
9.The results are presented in a descriptive manner but lack quantitative depth in places. For example, the magnitude of changes in gene expression and statistical metrics (e.g., p-values) should be included alongside visual data to support claims.
10.Over-reliance on Results Restatement: The discussion section reiterates many points from the results without providing additional interpretation or context. To avoid redundancy, the focus should be on the broader implications of the findings, rather than restating the data.
11.How do the results regarding the SNHG10/miR-665 axis align with or differ from previous studies on lncRNAs in ischemic stroke?
12.Although some limitations are acknowledged, they are not thoroughly explored. Additional limitations, such as the in vitro model's relevance to in vivo conditions and potential confounding factors in transfection efficiency, should be discussed.
13.Expanding on the translational potential of DHM and the next steps required to move towards clinical applications would strengthen the discussion.
14.The role of the SNHG10/miR-665/RASSF5 axis is a key focus of the paper, yet the molecular mechanisms underlying these interactions are not fully elaborated. A more in-depth discussion of possible signaling pathways and interactions would add significant value.
15.The conclusion section is relatively simplistic and does not adequately summarize the core findings and their scientific significance. It is recommended to expand the conclusion by emphasizing the key discoveries and their potential applications in stroke therapy.
16.For instance, there is a lack of further discussion regarding the experimental design, sample size, and animal studies. It is advised to provide a more detailed analysis of the study’s limitations.
17.While the experimental design is generally sound, the setup of experimental and control groups does not fully reflect scientific rigor, particularly in terms of the clinical relevance of the selected DHM concentrations, which requires further discussion.
18.Although the manuscript indicates that SNHG10 exerts its effects by regulating miR-665 and RASSF5, it does not sufficiently demonstrate that other potential pathways are not involved. Further investigation into the possibility of additional regulatory mechanisms is recommended.
19.The manuscript does not provide a detailed rationale for the selection of specific time points (e.g., 24 hours after OGD/R) for the experiments. Supplementing the manuscript with the justification for these choices is suggested.
20.There are certain errors or inadequacies in the grammar and sentence structure throughout the manuscript:
1)In the abstract, the sentence "In addition, the effects of the SNHG10/miR-665/RASSF5 axis on the activity, apoptosis, oxidative stress, and inflammation levels of SH-SY5Y cells were analyzed..." could be clearer. The word "levels" might create confusion, as it seems to refer to all listed terms. Consider rephrasing for clarity, such as “The effects of the SNHG10/miR-665/RASSF5 axis on SH-SY5Y cell activity, apoptosis, oxidative stress, and inflammatory markers were analyzed...”.
2)In the methods section, there are occasional shifts in tense, particularly between the present and past tenses. For example, “Cells were incubated...” (past tense) is followed by “RNA is isolated...” (present tense). To maintain consistency, it is recommended to use the past tense throughout the methods, as this is standard practice for describing completed experiments.
3)The phrase “SNHG10 has been reported to be a therapeutic target in various cancers, such as liver (7) and colorectal cancers (8)...” is awkward. It would be clearer to rephrase as, “...such as liver cancer (7) and colorectal cancer (8)...” to ensure proper parallelism and preposition use.
4)In the introduction, the sentence “IS pathology includes oxidative stress, neuroinflammation, and cell apoptosis (3).” includes redundant wording. "Cell apoptosis" is unnecessary because apoptosis refers specifically to cell death. It should be simplified to “IS pathology includes oxidative stress, neuroinflammation, and apoptosis.”
5)There are instances of comma splices, such as in “Cells were incubated in glucose-free DMEM and placed in an anaerobic chamber, they were then returned to normal culture conditions.” This should be corrected by either splitting into two sentences or using a conjunction: “...an anaerobic chamber. They were then returned to normal culture conditions.”
6)In the sentence "DHM reduced apoptosis, oxidative stress, and neuroinflammation, while improving viability of SH-SY5Y cells," the structure is not parallel. It would be better stated as, “DHM reduced apoptosis, oxidative stress, and neuroinflammation, while enhancing the viability of SH-SY5Y cells.”
Addressing these grammatical issues will improve the readability and scientific clarity of the manuscript.

Reviewer 3 ·

Basic reporting

This study investigated the neuroprotective effects of dihydromyricetin (DHM) against ischemic stroke (IS) in SH-SY5Y cells, focusing on the SNHG10/miR-665/RASSF5 axis and NF-κB signaling. DHM treatment attenuated OGD/R-induced cell damage by enhancing cell viability, reducing apoptosis, and mitigating oxidative stress and inflammation. Mechanistically, DHM suppressed SNHG10 expression, leading to increased miR-665 levels and subsequent inhibition of RASSF5. Further, DHM inactivation of NF-κB signaling contributed to its neuroprotective effects. The study provides novel insights into the protective mechanisms of DHM against IS, suggesting its potential as a therapeutic agent.
1. This research makes a significant contribution to understanding the potential therapeutic benefits of dihydromyricetin (DHM) in ischemic stroke (IS). The study's exploration of the SNHG10/miR-665/RASSF5/NF-κB pathway provides novel insights into the molecular mechanisms underlying DHM's neuroprotective effects. While the study provides compelling in vitro evidence, further research in vivo is necessary to validate these findings in animal models of IS. This would provide stronger evidence for DHM's therapeutic potential.
2. This abstract effectively conveys the study's key findings and provides a clear overview of the research. However, the abstract can be condensed by eliminating some of the less essential details and focusing on the most critical information. The phrase "novel insights" could be replaced with a more specific statement about the novelty of the findings, such as "this study provides the first evidence for..." or "this study identifies a novel mechanism...".
3. The introduction effectively sets the stage for the study by highlighting the significance of ischemic stroke (IS) and its complex pathophysiology, emphasizing the need for therapeutic strategies that target multiple pathological features. It then delves into the potential roles of long non-coding RNAs (lncRNAs), microRNAs (miRNAs), and the NF-κB pathway in IS, emphasizing the gaps in understanding, particularly regarding the interplay of these factors in the context of dihydromyricetin (DHM)'s potential neuroprotective effects. However, while the introduction effectively summarizes existing research, it could be strengthened by more explicitly connecting these pathways and highlighting their potential interactions in the context of IS. Furthermore, the introduction could be more concise by combining certain paragraphs and removing redundant information to improve the flow of information.
4. To enhance the reproducibility and reliability of your research, consider the following improvements in the Materials and Methods section:
1) For all critical reagents, such as antibodies and kits, include specific catalog numbers. This allows other researchers to easily locate and procure the same reagents, facilitating the replication of your experiment.
2) For key procedures like cell culture, transfection, RNA extraction, qPCR, and Western blotting, provide more comprehensive step-by-step instructions. Include specific details such as temperatures, incubation times, and reagent concentrations. This ensures that other researchers can accurately replicate the experiment.
3) Clearly state the models of instruments used in your experiments, such as centrifuges, incubators, and ELISA readers. This allows other researchers to replicate your methodology using comparable equipment.
4) For each experiment, clearly indicate the number of times it was repeated. For instance, "All experiments were repeated three times." This provides insight into the robustness of the results.
5. The mining evidence for the key influencing factors SNHG10/miR-665/RASSF5/NF-κ B in this study is insufficient, resulting in the overall design of the article appearing to be pre designed. Therefore, the author should strengthen the evidence chain supplementation of key factors SNHG10/miR-665/RASSF5/NF-κ B. In addition, the author needs to focus on elaborating on the results section rather than providing a general description.
6. The author indicated that “After validating the efficacy of the miR-665 mimic/inhibitor (Figure 4D), we found no significant change in the expression of SNHG10 (Figure 4E).” The author should explain why this result occurred.
7. The luciferase assay results in Figure 5M show that the miR-665 mimic reduces the luciferase activity of WT-RASSF5, suggesting a direct interaction. However, the lack of a significant difference between WT-RASSF5 and mut-RASSF5 cells co-transfected with the miR-665 inhibitor weakens this conclusion. This inconsistency requires further investigation and clarification.
8. The presented evidence in Figure 6 is not sufficient to conclusively support the conclusion that DHM improves SH-SY5Y cells via the SNHG10/miR-665/RASSF5 axis. The experiment involves co-transfection of SNHG10 cells with miR-665 mimic/si-RASSF5. This design makes it difficult to isolate the specific effect of DHM on the SNHG10/miR-665/RASSF5 axis. The observed changes could be due to the combined effects of the co-transfected elements rather than DHM's direct action on this axis.
9. The conclusion asserts that DHM "protected SH-SY5Y cells by reducing the transcription and translation of RASSF5 and inactivating the NF-κB signaling pathway through the SNHG10/miR-665 axis." However, the presented data only partially supports this statement. While the study shows DHM's protective effects and the modulation of the axis, it lacks direct evidence that DHM directly reduces RASSF5 transcription or translation.
10. The discussion mostly re-states the findings without critically analyzing their implications. It doesn't delve into the potential limitations of the study's design or explore the full implications of the findings for clinical practice. Some conclusions are not fully supported by the evidence presented, such as the claim that DHM protects against ischemic injury by modulating the SNHG10/miR-665/RASSF5 axis. While the study suggests a connection, it doesn't provide definitive evidence for this direct causal relationship.
11. The discussion repeatedly emphasizes the "novelty" of the findings without providing strong evidence to support its claims. The discussion acknowledges limitations but doesn't fully explore their potential impact on the findings. For example, the reliance on an in vitro model could significantly affect the generalizability of the findings.
12. The Figure legend should clearly and concisely describe the content of the figure, including what is being shown, the experimental conditions, and any statistical information.

Experimental design

The comment is merged into the Basic reporting

Validity of the findings

The comment is merged into the Basic reporting

---

## Round 0.2 · Minor Revisions

Your revised manuscript has been reviewed by three referees. While two reviewers recommend acceptance, one reviewer has raised some minor but important points that need to be addressed. Therefore, I invite you to submit a minor revision of your manuscript.

Please address the following points:
1. Add complete descriptions of materials, including company names and countries.
2. Clarify the experimental details regarding incubator usage.
3. Regarding the concern about GraphPad Prism's statistical capabilities, please include a brief justification for its use. You may wish to cite relevant publications that have employed GraphPad Prism for statistical analysis, which demonstrate that GraphPad Prism is widely accepted in the scientific community for both statistical analysis and graphical presentation.

We look forward to receiving your revised manuscript addressing these minor points.

·

Basic reporting

I believe that study has scientific importance.

Experimental design

pag. 98 - supplemented with 10% fetal bovine serum (Gibco "add the description")
pag. 142 - add the description: Dichloro-dihydro-fluorescein diacetate (DCFH-DA)
pag. 152-153 - how? incubators? more than one incubator were used?
pag. 155 - DAPI (4',6-diamidino-2-phenylindole)
Please, identify the name of the probes and company.

pag. 170 using a microplate reader (Thermo Fisher Scientific)
Please, identify the company and country of all materials used in the study

Validity of the findings

I still have doubts about the statistical veracity, because I consider GraphPad Prism a statistical program for developing ONLY graphics.

Reviewer 2 ·

Basic reporting

no comment

Experimental design

no comment

Validity of the findings

no comment

Additional comments

The revisions have greatly improved the manuscript. The clarity and structure are much better now.

Reviewer 3 ·

Basic reporting

The author's responses and revisions were specific and clear, supplementing the flaws and addressing my concerns.

Experimental design

no comment

Validity of the findings

no comment

---

## Round 0.3 · accepted · Accept

I have carefully reviewed your revised manuscript and the point-by-point responses to the reviewer comments. I am pleased to confirm that you have thoroughly addressed all the reviewers' concerns, particularly regarding the experimental details and statistical analysis methodology. The clarification about the use of SPSS for statistical analysis and GraphPad Prism for graphical representation is appropriate and strengthens the scientific rigor of your work. The addition of detailed information about reagents, materials, and experimental procedures enhances the reproducibility of your study. The current version meets our journal's standards for publication.